# Probiotic Supplementation Improves Gut Microbiota in Chronic Metabolic and Cardio-Cerebrovascular Diseases Among Chinese Adults over 60: Study Using Cross-Sectional and Longitudinal Cohorts

**DOI:** 10.3390/microorganisms13071507

**Published:** 2025-06-27

**Authors:** Xi Wang, Wanting Dong, Qiuying Liu, Xi Zeng, Yan Liu, Zheng Li, Yuanlong Pan, Qian Xiong, Na Lyu, Baoli Zhu

**Affiliations:** 1CAS Key Laboratory of Pathogenic Microbiology and Immunology, Institute of Microbiology, Chinese Academy of Sciences, Beijing 100101, China; 17862344536@163.com (X.W.);; 2University of Chinese Academy of Sciences, Beijing 100049, China; 3Beijing Children’s Hospital, Capital Medical University, Beijing 100069, China; 4Yingdong Intelligent Technology (Shandong) Co., Ltd., Jinan 250000, China; 5Department of Pathogenic Biology, School of Basic Medical Sciences, Southwest Medical University, Luzhou 646000, China; 6Jinan Microecological Biomedicine Shandong Laboratory, Jinan 250117, China; 7Beijing Key Laboratory of Antimicrobial Resistance and Pathogen Genomics, Beijing 100101, China

**Keywords:** probiotic supplementation, cross-sectional cohort, longitudinal cohort, gut microbiota, 16S rRNA gene sequencing

## Abstract

Probiotics demonstrate the ability to maintain intestinal homeostasis and promote gut health. However, their effects on gut microbiota in adults over 60 years old with chronic metabolic disease (CMD) or cardio-cerebrovascular disease (CCD) remain poorly understood. This study analyzed 1586 stool samples from 1377 adults (CMD, CCD, and healthy controls) using 16S rRNA sequencing. Cohort 1 (*n* = 1168) was used for cross-sectional analysis, while cohort 2 (*n* = 209) underwent longitudinal assessment over approximately 13 months. The results demonstrated that probiotics promoted significant gut microbiota alterations across both cohorts. Probiotic supplementation significantly increased lactobacilli in the CMD, CCD, and H groups. In both cohorts, probiotic supplementation enhanced *Butyricicoccus*, *Clostridium* sensu stricto 1, and *Coprococcus* in H groups, enhanced *Anaerostipes* and *Fusicatenibacter* in CMD groups, and reduced *Haemophilus* and *Lachnospira* in CCD groups. Notably, long-term supplementation not only elevated *Dorea*, *Eubacterium hallii* group, and *Blautia* in all groups but also suppressed *Klebsiella* and *Bilophila* in the CMD and CCD groups. Enterotype analysis revealed that probiotics increased the proportion of enterotype 1 and transition probabilities from enterotype 2 to 1 in the CMD and CCD groups, demonstrating that CCD/CMD gut microbiota exhibited greater responsiveness to probiotic modulation. Overall, this study suggests probiotics’ role in modulating adult gut microbiota and their potential benefits in chronic metabolic and cardio-cerebrovascular diseases.

## 1. Introduction

Globally, chronic metabolic disease (CMD) and cardio-cerebrovascular disease (CCD) have become significant public health challenges [1,2]. Growing evidence associates gut microbiota dysbiosis with the pathogenesis of both CMD and CCD [3,4,5]. Notably, patients with diabetes frequently display dysbiosis in gut microbial communities, marked by a significant reduction in *Akkermansia muciniphila* abundance, a microbial species recognized for its protective effects on improving intestinal barrier integrity, inhibiting inflammatory pathways, and modulating glucose homeostasis [3]. Conversely, patients with cardiovascular diseases often show abnormal proliferation of species such as *Prevotella copri*, *Massiliensis Ihubacter*, *Lachnoclostridium saccharolyticum*, and *Emergencia timonensis*, which contribute to elevated circulating trimethylamine N-oxide (TMAO) levels, thereby increasing cardiovascular risks [6,7,8]. Although CMD and CCD each have their own distinctive pathophysiological mechanisms, they frequently exhibit analogous risk factors and underlying pathological pathways, with the two groups of diseases being closely intertwined and demonstrating high comorbidity rates [9]. The synergistic interplay between immune dysfunction and gut microbiota dysbiosis further amplifies these pathological cascades [10], highlighting the potential of microbiota modulation as a novel preventive approach [11].

Current strategies for microbial community modulation in the gastrointestinal tract are predominantly centered on three validated improvement modalities, probiotic supplementation, prebiotic administration, and fecal microbiota transplantation (FMT), encompassing robust mechanistic rationales and clinical validations [12,13]. Prebiotics such as oligosaccharides, inulin derivatives, and fructooligosaccharide compounds preferentially enhance the growth and functional capacity of commensal microorganisms, thereby modulating the microbial community and promoting gastrointestinal homeostasis [14,15]. FMT entails the transplantation of fecal microbial consortia obtained from healthy donors into the intestinal environment of the recipients. The therapeutic efficacy of restoring gut microbiota homeostasis [13] has been evidenced by clinical trials reporting remission rates surpassing 79% in *Clostridioides difficile* infections (CDI) [16,17,18]. Among current strategies aimed at improving gut microbiota health, probiotic formulations are recognized as the most widely adopted consumable option, with their global market value exceeding USD 80 billion as of 2023 [19]. These preparations contain carefully selected live microbial strains that exert their effects through two primary mechanisms, colonization resistance and metabolic modulation [20]. Recent studies have documented their efficacy in optimizing gut microbiota structure, enhancing immune response, and promoting nutrient absorption [21,22,23]. However, there remain ongoing academic debates regarding the efficacy of probiotics [24], especially in light of clinical observations of suboptimal outcomes. For instance, *Bifidobacterium breve* BBG-001 failed to reduce the incidence of necrotizing enterocolitis and late-onset sepsis among preterm infants [25], while additional evidence suggests that some probiotic strains may delay microbiome reconstitution following antibiotic therapy [26]. Although these findings have not negated the usefulness of probiotics, they have sparked interest in precision therapies [27]. Notably, research frontiers have shifted from general health maintenance to targeted population applications, with elderly individuals (≥60 years old) emerging as a key focus due to their unique gut microbiota profiles [28,29]. As advancing age has been demonstrated to induce structural vulnerability in the gut microbiota of elderly individuals [30], and the high prevalence of chronic metabolic and cardio-cerebrovascular diseases further predisposes this population to dysbiosis [31], the modulatory effects of probiotic supplementation on gut microbiota in older adults with different health status is worthy of further research.

The current research on gut microbiota changes mediated by probiotics in patients over 60 years of age with either CMD or CCD is primarily focused on Western cohorts [32], whereas data from Asian populations, particularly Chinese cohorts, remain limited. This study employs 16S rRNA amplicon sequencing to characterize the modulatory effects of probiotic supplementation on gut microbiota structure in Chinese individuals over 60 years old under varying health conditions.

## 2. Materials and Methods

### 2.1. Probiotic Strains

The Basegar brand probiotic (manufactured by Yingdong Intelligent Technology (Shandong) Co., Ltd., Jinan, Shandong Province, China) contains *Lacticaseibacillus paracasei* Lpc-37, *Lacticaseibacillus rhamnosus* HN001, *Bifidobacterium animalis* subsp. *lactis* HN019, *Lactobacillus acidophilus* NCFM, and *Lacticaseibacillus rhamnosus* Lr-32. The concentration of bacterial strains in the probiotic supplements remains undisclosed.

### 2.2. Sample Collection

A total of 3926 stool samples were collected from Chinese adults between November 2022 and September 2024 through Yingdong Intelligent Technology (Shandong) Co., Ltd., with sampling sites spanning Shandong Province, Beijing, Anhui Province, Tianjin, Jiangxi Province, and Hebei Province. All participants completed structured questionnaires collecting demographic parameters (age, gender, height, weight, residential region), lifestyle characteristics (smoking status, alcohol consumption, dietary patterns), medical history (intestinal versus other medical history) and medication records (probiotic supplementation, antibiotics use, and other therapeutic regimens; no dosage information available). Participant screening was conducted based on questionnaire information using the following inclusion criteria: (1) aged 60–90 years; (2) participants with a self-reported history of chronic metabolic disease (including hypertension, hyperlipidemia, diabetes mellitus, or gout; conditions were self-reported, with participants taking medications for non-corresponding disease categories excluded, and no clinical medical records provided) or cardio-cerebrovascular disease (including premature cardiac contractions, coronary heart disease, mitral valve insufficiency, atrial fibrillation, myocardial infarction, angina pectoris, heart failure, myocardial ischemia, cerebral infarction, cerebellar atrophy, and cerebral hypoperfusion; conditions were self-reported, with participants taking medications for non-corresponding disease categories excluded, and no clinical medical records provided) as well as healthy individuals (self-reported as healthy and not taking medications for psychiatric, neurological, hepatic, renal, cholecystic, intestinal, gastric, cardio-cerebrovascular diseases, cancer, or metabolic disorders; conditions were self-reported, with participants taking medications for non-corresponding disease categories excluded, and no clinical medical records provided); (3) participants who had maintained probiotic supplementation for over 8 weeks qualified for enrollment in the probiotics group (with regard to daily dosage, participants voluntarily self-administered the supplements without clinical supervision, while the duration of intake was self-reported by participants). Exclusion criteria encompassed the following conditions: (1) recent antibiotic usage (≤30 days before enrollment); (2) documented malignancies or chronic pathologies affecting gastrointestinal, hepatobiliary, pulmonary, renal, psychiatric, or neurological systems. Prior to study commencement, written informed consent was acquired from all subjects following established protocols. A total of 1377 participants contributed 1586 fecal samples retained for downstream analytical processes. From this population, 209 subjects provided longitudinal samples through sequential collection phases: baseline specimens during the fourth quarter of 2022 (October–December) followed by subsequent sampling in the first quarter of 2024 (January–March). Single fecal sample collection was implemented for the remaining 1168 participants.

### 2.3. Study Cohorts

In order to investigate the effects of probiotic supplementation on gut microbiota under different health statuses, we established cohort 1, comprising 1168 participants aged over 60 years who provided single fecal samples. This cohort was grouped according to the following characteristics: (1) health status (chronic metabolic disease, cardio-cerebrovascular disease, and healthy controls) and (2) probiotic supplementation (the probiotic supplementation group and the no probiotic supplementation group) for subsequent cross-sectional analysis.

To examine the long-term effects of probiotic supplementation on gut microbiota across individuals with different health status, we established cohort 2, comprising 418 longitudinal fecal samples collected from 209 subjects. Cohort 2 included 209 participants aged over 60 years undergoing continuous probiotic supplementation. Two sampling rounds were conducted in 2022 and 2024, with an average sampling interval of 13.77 months between rounds. The observed intervals were distributed as follows: 12 months (12 cases), 13 months (28 cases), 14 months (167 cases), 15 months (1 case), and 16 months (1 case). Samples in cohort 2 were categorized by health status (chronic metabolic disease, cardio-cerebrovascular disease, and healthy controls) and collection time (2022 and 2024 batches) to enable longitudinal analyses.

### 2.4. DNA Extraction and Sequencing

The fecal specimens underwent bacterial genomic DNA isolation using the QIAamp PowerFecal DNA Kit (manufactured by Qiagen N.V., Hilden, Germany) following standardized operational protocols. The V3–V4 regions of the bacterial 16S rRNA genes were amplified using the 341F (5′-CCTACGGGNBGCASCAG-3′) and 805R (5′-GACTACNVGGGTATCTAATCC-3′) primers. Sequencing libraries were subjected to high-throughput analysis through the Illumina MiSeq system (manufactured by Illumina, Inc., San Diego, CA, USA), generating 250 bp paired-end reads.

### 2.5. Bioinformatics Analysis

Raw data underwent processing through QIIME2 (v2024.2) [33]. DADA2 was employed to generate amplicon sequence variants (ASVs) through unique sequence grouping [34]. Taxonomic classification was executed via the q2-feature-classifier plugin [35], utilizing the SILVA database [36] with a classify-sklearn naïve Bayes taxonomic classifier. Microbial community functional profiles were predicted using PICRUSt2 software (v2.5.2) [37], referencing the KEGG database. Enterotype clustering at genus level followed established methodology [38], where dominant genera determined classification: *Bacteroides* primarily characterized enterotype1 (E1), while *Prevotella* predominantly defined enterotype2 (E2). Cohort 2 samples were divided into two temporal categories corresponding to 13-month intervals, with individual enterotype assignments based on modal occurrence within each period. Markov chain analysis implemented through a published R script (v4.4.0) [39] quantified and visualized inter-enterotype transition dynamics. Since 2020, under updated taxonomy, “*Lactobacillus*” and “*Bifidobacterium*” have been revised to “lactobacilli” and “bifidobacteria”, respectively [40,41], except for specific species names.

### 2.6. Statistical Analysis

Diversity assessments at ASV levels were performed employing the vegan package [42]. The Chao1 index served to estimate α-diversity, complemented by Shannon index calculations, whereas β-diversity evaluation utilized Bray–Curtis dissimilarity matrices. Multivariate statistical comparisons in principal coordinate analyses (PCoAs) were conducted through weighted permutational multivariate analysis of variance tests (PERMANOVA/adonis). Bacterial abundance variations across groups were detected using linear discriminant analysis effect size (LEfSe) methodology. STAMP software (v2.1.3) facilitated identification of significant KEGG pathway discrepancies (via PICRUSt2) through Wilcoxon tests, incorporating false discovery rate (FDR)-adjusted *p*-values. Intergroup comparisons of microbial relative abundance employed nonparametric Wilcoxon rank-sum tests.

## 3. Results

### 3.1. Characteristics of the Cohorts

Cohort 1 comprised 1168 participants aged over 60 years. Based on health status obtained from questionnaire reports, participants were divided into three groups, the chronic metabolic disease group (CMD group, *n* = 547), the cardio-cerebrovascular disease group (CCD group, *n* = 339), and healthy group (H group, *n* = 282). Further division by probiotic supplementation status yielded two groups: the probiotic supplementation group (P group, *n* = 660) and the no probiotic supplementation group (NP group, *n* = 508). Utilizing a 2 × 3 factorial design combining health status and probiotic supplementation status, six research subgroups were established, including chronic metabolic disease group without probiotic supplementation (CMD-NP group, *n* = 212), chronic metabolic disease group with probiotic supplementation (CMD-P group, *n* = 335), cardio-cerebrovascular disease group without probiotic supplementation (CCD-NP group, *n* = 164), cardio-cerebrovascular disease group with probiotic supplementation (CCD-P group, *n* = 175), healthy group without probiotic supplementation (H-NP group, *n* = 132), and healthy group with probiotic supplementation (H-P group, *n* = 150). Detailed grouping information for cohort 1 is shown in Table 1.

Cohort 2 included 209 subjects aged over 60 years undergoing long-term probiotic supplementation. Participants with long-term probiotic supplementation were divided into three groups based on health status, comprising the CMD-P group (*n* = 83), the CCD-P group (*n* = 76), and the H-P group (*n* = 50). All participants provided fecal samples during two collections in 2022 and 2024 with an average sampling interval of 13.77 months, yielding a total of 418 samples. The samples were temporally divided into two batches: the 2022 batch (October–December, *n* = 209) and the 2024 batch (January–March, *n* = 209). Six subgroups were subsequently established in cohort 2 based on health status, probiotic supplementation status, and sampling time, including the 2022 batch of CMD-P group samples (2022-CMD-P group, *n* = 83), the 2024 batch of CMD-P group samples (2024-CMD-P group, *n* = 83), the 2022 batch of CCD-P group samples (2022-CCD-P group, *n* = 76), the 2024 batch of CCD-P group samples (2024-CCD-P group, *n* = 76), the 2022 batch of H-P group samples (2022-H-P group, *n* = 50), and the 2024 batch of H-P group samples (2024-H-P group, *n* = 50). Detailed sample information for cohort 2 is presented in Table 2.

### 3.2. Patients with CMD or CCD Exhibit Differences in Gut Microbiota Composition Compared to Healthy Controls

Through comparative analysis of gut microbiota among the three groups without probiotic supplementation in cohort 1, this study compared microbial communities between the CMD and CCD groups (CMD-NP and CCD-NP) and the healthy controls (H-NP) in participants aged over 60 years. α-Diversity analysis demonstrated significantly higher Chao1 and Shannon indices in both the CCD-NP and H-NP groups relative to the CMD-NP group (*p* < 0.05). Although the H-NP group exhibited a trend toward higher Chao1 and Shannon indices compared to the CCD-NP group, the intergroup differences were not statistically significant (*p* > 0.05; Figure 1A). β-Diversity was evaluated by using PCoA based on Bray–Curtis distance in the NP groups. Adonis tests revealed significant differences in gut microbiota structure between individuals with CMD or CCD and healthy controls (*p* < 0.05; Figure 1B). Taxonomic analyses at the genus level identified *Bacteroides*, *Prevotella*, *Faecalibacterium*, *Parabacteroides*, *Escherichia Shigella*, *Lachnospira*, *Blautia*, bifidobacteria, *Subdoligranulum*, and *Sutterella* as the most abundant gut microbial genera across all groups (Figure 1C). We next applied LEfSe to identify differentially abundant bacteria between the CMD and CCD groups (CMD-NP and CCD-NP groups) and the H-NP group. This investigation sought to identify microbial patterns linked to CMD, CCD, and healthy states. Compared to the H-NP controls, the CMD-NP subjects demonstrated significantly higher abundances of *Veillonella*, *Klebsiella*, and *Escherichia Shigella*, and showed notable depletion of multiple taxa in the gut microbiota, including *Eubacterium coprostanoligenes* group, *Eubacterium siraeum* group, *Eubacterium ventriosum* group, Oscillospiraceae UCG 002, *Anaerostipes*, Oscillospiraceae UCG 003, *Akkermansia*, and bifidobacteria (Figure 1D). The CCD-NP subjects exhibited microbial enrichment of *Lachnoclostridium*, *Klebsiella*, *Veillonella*, and *Escherichia Shigella* relative to the H-NP controls, contrasting with diminished populations of *Eubacterium ventriosum* group, Lachnospiraceae NK4A136 group, *Ruminococcus*, and *Flavonifractor* in the CCD-NP gut microbiota (Figure 1D).

### 3.3. Probiotic Supplementation Promotes Abundance of Beneficial Bacteria in the CMD and CCD Groups

By comparing the gut microbiota of the three health status categories with and without probiotic supplementation in cohort 1, we elucidated the effects of probiotics on intestinal microbial communities in both populations with and without CMD/CCD aged over 60 years. Analysis of α-diversity demonstrated reduced Chao1 richness indices across P groups in three health status when compared to their NP counterparts (*p* < 0.05; Figure 2A). While microbial diversity assessed through Shannon indices exhibited upward trends in all P groups relative to their NP counterparts, these variations failed to reach statistical significance (*p* > 0.05; Figure 2A). Bray–Curtis distance, a widely used metric for assessing gut microbiota community similarity, revealed significantly greater structural differences between the CMD and CCD groups without probiotic supplementation and the H-NP group compared to that between the CMD and CCD groups with probiotic supplementation and the H-P group (*p* < 0.05; Figure 2B). These results suggest probiotic supplementation diminishes structural dissimilarities in intestinal microbiota between the CMD and CCD groups (CMD-P and CCD-P) and their healthy counterparts (H-P) (*p* < 0.05; Figure 2B). PCoA analysis was employed to assess the effects of probiotic supplementation on β-diversity in the three health status categories. The adonis test results demonstrated that probiotic supplementation significantly altered gut microbiota structure in the CMD and CCD groups and the healthy populations (*p* < 0.05; Figure 2C). LEfSe analysis was employed to identify gut bacterial taxa specifically associated with probiotic supplementation within each health status category (Figure 2D). After comparing the differences between P and NP groups within each of the three health status categories, it was found that probiotic supplementation led to a significant increase in bacteria such as *Coprococcus*, *Clostridium* sensu stricto 1, lactobacilli, and *Butyricicoccus* in the H-P group compared to the H-NP group. Additionally, genera such as *Megamonas*, *Megasphaera*, *Haemophilus*, *Fusobacterium*, *Desulfovibrio*, *Phascolarctobacterium*, *and Prevotella* were significantly less abundant in the H-P group compared to the H-NP group (Figure 2D). Probiotic supplementation resulted in a significant enrichment of bacteria such as *Eubacterium siraeum* group, *Alloprevotella*, *Fusicatenibacter*, *Anaerostipes*, lactobacilli, and bifidobacteria in the CMD-P group compared to the CMD-NP group, whereas genera such as Lachnospiraceae UCG 003, *Fusobacterium*, *Ruminococcus gnavus* group, *Eubacterium ruminantium* group, *Megamonas*, Prevotellaceae NK3B31 group, and *Prevotella* were significantly reduced (Figure 2D). In the CCD-P group, probiotic supplementation resulted in significant enrichment of bacteria such as *Achromobacter*, *Holdemanella*, *Alloprevotella*, *Oscillibacter*, *Flavonifractor*, *Alistipes*, lactobacilli, and bifidobacteria compared to the CCD-NP group, while Erysipelotrichaceae UCG 003, *Desulfovibrio*, *Haemophilus*, *Fusobacterium*, *Megamonas*, Prevotellaceae NK3B31 group, *Phascolarctobacterium*, *Lachnospira*, *Faecalibacterium*, and *Prevotella* were significantly reduced (Figure 2D).

To investigate the effects of probiotic supplementation on gut bacteria associated with diseases and health in individuals aged over 60 years, we integrated both the differential bacterial taxa between disease and healthy groups without probiotic supplementation and the differential bacterial taxa between P and NP groups within each health status group in cohort 1. Statistical analyses were performed on bacterial taxa from both integrated comparative sets. Probiotic supplementation significantly reduced microbial taxa including *Megamonas*, *Fusobacterium*, and *Prevotella* across three health status categories (*p* < 0.05; Figure 3A). The CMD-P and CCD-P groups showed significant suppression of Prevotellaceae NK3B31 group (*p* < 0.05; Figure 3A), while *Desulfovibrio*, *Haemophilus*, and *Phascolarctobacterium* were significantly diminished in CCD-P and H-P groups (*p* < 0.05; Figure 3A). Probiotic supplementation demonstrated statistically significant enrichment of lactobacilli across three health status groups (*p* < 0.05; Figure 3B), with concurrent increases in *Alloprevotella* and bifidobacteria specifically observed in CMD-P and CCD-P groups (*p* < 0.05; Figure 3B). Comparative analysis revealed three microbial taxa displaying significant abundance differences between the CMD and CCD groups without probiotic supplementation (CMD-NP and CCD-NP) and H-NP controls. Probiotic supplementation in the CMD and CCD groups (CMD-P and CCD-P) shifted microbial profiles toward H-NP controls, whereas the CMD and CCD groups without probiotic supplementation maintained distinct trends. Bifidobacteria and *Anaerostipes* exhibited significantly lower abundance in CMD-NP compared to both H-NP and CMD-P groups (*p* < 0.05; Figure 3B,C), while *Flavonifractor* in CCD-NP was significantly reduced relative to H-NP and CCD-P groups (*p* < 0.05; Figure 3C). These findings suggest probiotics promote the enrichment of potentially beneficial taxa in CMD (bifidobacteria and *Anaerostipes*) and CCD (*Flavonifractor*).

### 3.4. Long-Term Probiotic Supplementation Inhibits Gut Pathobionts in the CMD and CCD Groups While Increases the Abundance of Beneficial Bacteria

The longitudinal analysis of gut microbiota differences in cohort 2 within each of the three health status categories revealed the effect of long-term probiotic supplementation on both CMD and CCD and healthy populations aged over 60 years. The α-diversity analysis showed that the long-term probiotic supplementation reduced the Chao1 index of gut microbiota across three health status categories, while the Shannon index exhibited a slight increase. However, neither change reached statistical significance (*p* > 0.05; Figure 4A). PCoA revealed that long-term probiotic supplementation significantly altered the gut microbiota structure in different groups (including both populations with and without CMD and CCD; adonis test, *p* < 0.05; Figure 4B). LEfSe analysis was employed to assess the specific effects of long-term probiotic supplementation on the gut microbiota in the three health status categories (Figure 4C). Compared with the gut microbiota of the 2022 batch, long-term probiotic supplementation resulted in a significant reduction in bacteria including *Butyricimonas*, *Klebsiella*, Lachnospiraceae NK4A136 group, *Romboutsia*, *Alistipes*, and *Akkermansia*, as well as a significant enrichment of bacteria such as lactobacilli, *Butyricicoccus*, *Clostridium* sensu stricto 1, *Enterococcus*, *Dorea*, *Eubacterium hallii* group, *Coprococcus*, *Fusicatenibacter*, and *Blautia* in the 2024-H-P group (Figure 4C). Compared to the 2022 batch, long-term probiotic supplementation in the 2024-CMD-P group resulted in a marked decrease in the abundance of *Bilophila*, *Klebsiella*, *Phascolarctobacterium*, and *Sutterella*, alongside an increase in *Butyricicoccus*, Lachnospiraceae ND3007 group, *Coprococcus*, *Streptococcus*, *Anaerostipes*, lactobacilli, *Ruminococcus*, *Agathobacter*, *Romboutsia*, *Dorea Eubacterium hallii* group, *Fusicatenibacter*, and *Blautia* (Figure 4C). In the 2024-CCD-P group, long-term probiotic supplementation, compared to the 2022 batch, was associated with reduced levels of *Haemophilus*, *Bilophila*, *Klebsiella*, and *Lachnospira*, and it was found to significantly enrich the gut microbiota with lactobacilli, *Butyricicoccus*, *Clostridium* sensu stricto 1, Oscillospiraceae NK4A214 group, *Eubacterium hallii* group, *Fusicatenibacter*, *Dorea*, and *Blautia* (Figure 4C).

Longitudinal analysis of different health status groups in cohort 2 revealed consistent microbial modulation following extended probiotic supplementation. Long-term probiotic supplementation significantly enriched lactobacilli, *Dorea*, *Eubacterium hallii* group, *Fusicatenibacter*, and *Blautia* across three health status categories (*p* < 0.05; Figure 5A). Long-term probiotic supplementation promoted a significant increase in the abundance of *Butyricicoccus* and *Coprococcus* in chronic metabolic disease patients and healthy populations (*p* < 0.05; Figure 5A). Conversely, *Klebsiella* and *Bilophila* demonstrated a significant reduction in the CMD and CCD groups (*p* < 0.05; Figure 5B).

### 3.5. The Modulatory Effect of Probiotic Supplementation on Gut Enterotype in the CMD and CCD Groups than in the Healthy Group

We performed enterotype analysis on cohort 1 and cohort 2 to investigate the effect of probiotics on enterotype distribution and transformation in individuals over 60 years old with varying health status. The enterotype analysis of cohort 1 revealed two distinct clusters: enterotype 1 (E1), predominantly characterized by *Bacteroides*; and enterotype 2 (E2), primarily dominated by *Prevotella* (Figure 6A). Comparison of enterotype proportions across groups in cohort 1 revealed that E1 was more prevalent than E2. Notably, probiotic supplementation promoted an increase in the proportion of E1 within the gut microbiota of both CMD and CCD patient groups, while healthy individuals demonstrated a contrasting trend (Figure 6B,C). Chi-square analysis revealed no significant difference in E1 proportions between three health status categories with probiotic supplementation and their corresponding groups without probiotic supplementation (*p* > 0.05). However, within the P groups, the CMD and CCD groups (CCD-P and CMD-P) demonstrated significantly higher E1 proportions compared to the H-P (*p* < 0.05; Appendix A). Enterotype analysis of samples from cohort 2 identified two distinct enterotypes (E1 and E2), consistent with the findings in cohort 1 (Figure 6D). Comparison of enterotype proportions across groups in cohort 2 showed that E1 was more prevalent than E2. Furthermore, long-term probiotic supplementation was associated with a slight increase in E1 proportions within all health status groups of cohort 2 (Figure 6F). To assess enterotype transformation before and after more than 13 months of probiotic supplementation, we conducted a Markov chain analysis. Notably, the CMD-P (36.84%) and CCD-P (31.82%) groups exhibited significantly higher conversion rates from E2 to E1 compared to the H-P group (20.00%).

## 4. Discussion

Comparative analysis of gut microbiota structure among three health status categories without probiotic supplementation revealed significantly higher gut microbiota richness and diversity in the healthy group (H-NP) compared to the CMD-NP group, consistent with previous studies [43,44]. Notably, the CCD-NP group exhibited comparable gut microbiota diversity and richness to the H-NP group, while demonstrating significantly higher α-diversity indices than those of the CMD-NP group. This observation might result from pharmaceutical regimen heterogeneity and health condition variations among geriatric patients, which could enhance interpersonal microbial community variations [45], possibly accounting for the anomalous elevation in the CCD-NP group’s mean α-diversity indices. The β-diversity analysis revealed significant gut microbiota differences among the three health status groups, both without and with probiotic supplementation (Appendix A). This indicates that probiotic supplementation cannot eliminate disease-associated differences in gut microbiota structure. Comparative analysis of gut microbiota between CMD and CCD groups without probiotic supplementation and healthy controls without probiotic supplementation revealed disease-associated opportunistic pathogens and health-associated potentially beneficial bacteria. Notably, opportunistic pathogens including *Veillonella*, *Klebsiella*, and *Escherichia Shigella* demonstrated significant enrichment in both CMD-NP and CCD-NP groups relative to the H-NP group. These findings align with prior studies that have highlighted the prevalence of these bacteria in chronic metabolic and cardiovascular diseases [46,47,48,49,50]. Additionally, *Lachnoclostridium* was significantly enriched in the CCD-NP group compared to the H-NP group. Available evidence suggests a significant correlation between the abundance of *Lachnoclostridium* and the development of atherosclerosis [8], as well as its association with diseases such as obesity, hypertension, and diabetes [51,52,53]. Conversely, the *Eubacterium ventriosum* group, a known short-chain fatty acid (SCFA)-producing bacterium associated with ameliorating hyperglycemia and regulating lipid metabolism, showed significantly lower abundance in the CMD-NP and CCD-NP groups compared to the H-NP group [54,55].

Probiotic supplementation significantly reduced gut microbiota richness within each of the three health status categories in cohort 1, while a slight increase in diversity was observed, though this was not statistically significant. The reduced gut microbiota richness in cohort 1 might be associated with the competitive exclusion effect of exogenous probiotics, which suppressed the growth of other indigenous bacteria [56]. Although gut microbiota richness is not an absolute health indicator, numerous studies have shown that higher richness is generally linked to a more stable gut microbiota structure and stronger host resistance [57]. Based on these observations and analyses, we proposed that probiotics may not confer universal benefits to all individuals [58,59]. This finding highlighted the need for a more cautious approach to probiotic applications. The effect of long-term probiotic supplementation (cohort 2) on α-diversity aligns with the trend observed in cohort 1. The difference was that the decrease in the gut microbiota richness caused by long-term probiotic supplementation showed no significant difference. Since cohort 1 compared gut microbiota differences between individuals with and without probiotic supplementation, while cohort 2 examined longitudinal gut microbiota differences within the same group before and after long-term supplementation, we hypothesize that the observed changes in richness index may result from certain gut microbiota features becoming stabilized during long-term probiotic supplementation, thereby reducing the extent of changes in these microbial characteristics [60]. PCoA revealed that probiotic supplementation (cohort 1) was associated with significant structural changes in gut microbiota within each of the three health status groups. A subsequent comparison of the Bray–Curtis dissimilarity between the four subgroups with CMD or CCD and the two healthy subgroups revealed that probiotic supplementation resulted in a more similar gut microbiota structure between the CCD-P and CMD-P groups and the H-P groups. Combined with the aforementioned β-diversity findings, the results indicate that while probiotic supplementation promoted convergence of gut microbiota structure between the CCD/CMD and healthy groups, it did not restore patients’ microbiota to the baseline levels observed in healthy populations [61]. Following long-term probiotic supplementation (cohort 2), PCoA also revealed significant structural alterations in gut microbiota within each of the three health status groups, consistent with the structural changes observed in cohort 1. This study compared the commonalities and differences in gut microbiota alterations between matched health status categories in cohort 1 (groups with probiotic supplementation versus those without) and cohort 2 (before and after long-term supplementation), based on differences in probiotic supplementation modalities. To identify cross-cohort commonality, we characterized the overlap between the differential bacterial taxa in the three health status categories of cohort 1 under probiotic supplementation and those in cohort 2 following long-term probiotic supplementation. Probiotic supplementation significantly enriched lactobacilli within each of the three health status categories in both cohorts [62]. In the H group, the relative abundance of *Butyricicoccus*, *Clostridium* sensu stricto 1, and *Coprococcus* increased significantly. In contrast, the CMD groups exhibited notable enrichments in *Anaerostipes* and *Fusicatenibacter*. Concurrently, the CCD groups experienced significant reductions in *Haemophilus* and *Lachnospira* following probiotic supplementation. Consistent with previous studies, long-term probiotic supplementation has been demonstrated to significantly enhance the abundance of *Fusicatenibacter* and lactobacilli in the gut of healthy individuals [63] (Appendix A). A growing body of research has identified bacteria such as *Butyricicoccus*, *Coprococcus*, and *Fusicatenibacter* as key SCFA producers, which play a crucial role in gut health. These bacteria are often depleted in individuals with diabetes and have been associated with improved lipid metabolism [64,65,66,67,68,69,70,71]. *Clostridium* sensu stricto 1 is also capable of producing large amounts of SCFAs, but its relevance to human diseases remains poorly understood [72,73,74]. High levels of SCFA are inversely associated with conditions such as hypertension, cardiovascular disease, stroke, obesity, and diabetes [75,76,77,78,79,80,81], suggesting that the aforementioned SCFA-producing bacteria may exert beneficial effects on human health. Conversely, *Haemophilus* and *Lachnospira*, which are identified as potential pathogenic risk factors, were suppressed following probiotic supplementation. *Haemophilus* has also been associated with diabetic nephropathy and obesity [82,83]. However, current evidence supporting an association between *Haemophilus* and cardiovascular disease remains limited. There are reports of increased relative abundance of *Lachnospira* in the gut of patients with long-term type 2 diabetes (T2D) and obesity [84,85]. A noteworthy bacterium, *Anaerostipes*, exhibited a significant reduction in the CMD-NP group compared to the H-NP group. Furthermore, probiotic supplementation prompted a significant increase in *Anaerostips* in both cohorts of the CMD group. Among all bacteria examined in this study, it was the only bacterium associated with health and steadily increased by probiotic supplementation in the CMD group. *Anaerostipes* is a key genus of important butyric acid-producing bacteria in the gut, playing a crucial role in host metabolic health. Notably, its abundance is significantly reduced in patients with diabetes [86].

In addition to the microbial changes shared between the two cohorts, we identified unique gut microbiota alterations in cohort 2 that were specifically associated with long-term probiotic supplementation. Long-term probiotic supplementation significantly enriched *Dorea*, *Eubacterium hallii* group, and *Blautia* within each of the three health status categories, while concurrently reducing the abundance of *Klebsiella* and *Bilophila* in the CMD and CCD groups. The research evidence indicates that gut microbiota, including the *Eubacterium hallii* group, *Dorea*, and *Blautia*, play crucial roles in improving lipid metabolism and enhancing blood glucose homeostasis [64,65,69,87,88,89,90,91]. Previous studies have reported a significant increase in *Klebsiella* abundance in patients with T2D, with metformin treatment significantly decreasing its gut abundance [48]. The present study demonstrated that long-term probiotic supplementation led to a significant decrease in *Klebsiella* abundance in all three health status categories (Appendix A). It is worth mentioning that *Klebsiella* was co-enriched in the CMD-NP and CCD-NP groups of cohort 1 compared to the H-NP group. *Bilophila* has been identified as a potential new risk factor for diet-induced hepatic steatosis in humans, and its reduction is beneficial in ameliorating atherosclerosis in high-fat diet hosts [92].

Given that probiotic supplementation promoted structural convergence of gut microbiota across groups, we conducted enterotype analysis to characterize the differential modulation of gut microbiota by probiotic supplementation between disease and healthy groups. The gut microbiota of participants across both cohorts were identified to comprise two enterotypes: enterotype 1 (E1, *Bacteroides*-dominant) and enterotype 2 (E2, *Prevotella*-dominant) [93]. Comparative analysis revealed that the prevalence of E1 was markedly higher than that of E2 in all groups across both cohorts. Probiotic supplementation (cohort1) increased E1 prevalence in the gut microbiota of the CMD and CCD populations but decreased it in the healthy (H) population. The chi-square test demonstrated that probiotic supplementation exerted a significantly greater effect on the E1 distribution of gut microbiota in the CCD-P and CMD-P groups than in the H-P group (*p* < 0.05). Additionally, we found that probiotic supplementation suppressed the growth of *Prevotella*. Analysis of *Prevotella* at the species level revealed that probiotic supplementation resulted in significant reductions in the relative abundance of *Prevotella copri* in both the CCD and H groups, as well as *Prevotella stercorea* in all three health status categories (Appendix A). A significant increase in *Prevotella copri* abundance has been strongly associated with cardiovascular disease [7]. Furthermore, emerging evidence indicates that *Prevotella stercorea* exhibits significant positive correlation with key metabolic parameters, including body mass index (BMI), fasting blood glucose, and visceral fat mass [94]. The role of *Prevotella* in the gut is highly complex, with its functions influenced by host diet, strain characteristics, and interactions with other microorganisms [95,96,97]. Although certain *Prevotella* strains exhibit potential pro-inflammatory and pathogenic effects [98,99], their beneficial functions in metabolic health and immune regulation should not be overlooked [100]. Future research needs to further explore the functional mechanisms of different *Prevotella* strains to better understand their roles in health and disease. We further analysed the effect of long-term probiotic supplementation (cohort 2) on the transition probability of enterotypes in each group using Markov chains and found that long-term probiotics use promoted the conversion of gut microbiota from E2 to E1 in different groups of participants. Furthermore, the conversion rate to E1 was higher in the CMD and CCD groups, each compared to the healthy group. Enterotype analysis demonstrated that probiotic supplementation exerted more significant microbiota modulating effects in populations with CMD or CCD than in healthy individuals. This differential response likely stems from the frequent dysbiosis state of patients’ gut microbiota, whereas healthy subjects exhibit stronger resistance to exogenous probiotic colonization owing to their higher homeostatic stability of gut microbiota structure. In a study by Gu et al., an intervention with acarbose in type 2 diabetes showed a more significant improvement in metabolic parameters among enterotype 1 diabetes patients than in enterotype 2 patients [101]. A study conducted in Asian children found that a reduction in bifidobacteria abundance in the gut was associated with a greater degree of obesity in participants of the E1 enterotype [102]. It appears challenging to modify enterotype solely through dietary interventions [103,104,105]. This study has demonstrated that long-term probiotic supplementation facilitates the transition of gut microbiota from E2 to E1. However, the issue of whether this microbial shift confers long-term health benefits requires further investigation. Each of the two enterotypes exhibits unique digestive functions [106]. We therefore employed the PICRUSt2 tool to predict metabolic functions of the gut microbiota across groups in cohort 2. PICRUSt2 predictions indicated that long-term probiotic supplementation stimulated energy metabolism processes, including galactose metabolism and glycolysis/gluconeogenesis, in all three health status groups (Appendix A). This observation suggests that long-term probiotic supplementation may enhance the efficiency of glucose conversion into energy and contribute to maintaining stable blood glucose levels [107,108]. Furthermore, long-term probiotic supplementation was shown to promote metabolic processes, including amino acids biosynthesis and peptidoglycan biosynthesis, in the CMD and CCD groups (Appendix A). Previous studies have shown that enterotype 2 is characterized by hydrolytic enzymes that effectively degrade plant fibers but have limited capacity for lipolytic and protein hydrolytic fermentation. Conversely, enterotype 1 possesses enzymatic systems adapted for the breakdown of zoological polysaccharides, demonstrating elevated capacities for glycolytic processing and protein breakdown [109,110,111], consistent with the functional prediction results from our PICRUSt2 analyses. Emerging evidence demonstrates that the increased abundance of the *Bacteroides* exhibits a negative association with age in individuals maintaining metabolic health beyond 40 years [112,113]. Nevertheless, the health effects of gut microbiota shifting toward a *Bacteroides*-dominant enterotype (E1) in elderly patients with chronic metabolic or cardio-cerebrovascular disease still require further research.

In conclusion, probiotic supplementation improves gut microbiota composition across participant subgroups, with long-term use exerting significantly stronger modulatory effects in both CMD and CCD populations compared to healthy individuals. However, several limitations warrant acknowledgment: (1) as an observational study, residual confounding from unquantified dietary components and medication dosages/durations persists despite baseline adjustment; (2) probiotic supplementation adherence was assessed through purchase interval data, lacking monitoring of dosage regimens or actual daily intake; (3) diet and medication data captured categorical patterns without quantitative precision; (4) intergroup sample size imbalance may introduce statistical bias in gut microbiota comparative analyses. Future investigations should establish multi-center cohorts with standardized dynamic monitoring frameworks for dietary intake, pharmacotherapy, and lifestyle factors, and design targeted experiments to delineate the time-dependent effects of probiotic intervention on microbiota dynamics across disease subtypes.

## 5. Conclusions

This study investigated the modulatory effects of probiotics on gut microbiota in older adults with distinct health status. The β-diversity analyses of cohort 1 and cohort 2 demonstrated that probiotics significantly altered the gut microbiota structure in all three health status groups. Bray–Curtis dissimilarity and adonis analyses in cohort 1 revealed that, while probiotic supplementation promoted the structural convergence of gut microbiota between the CMD/CCD and healthy groups, it could not restore patients’ microbiota structure to baseline levels observed in healthy populations. In both cohorts, probiotic supplementation significantly increased the abundance of lactobacilli within each of the three health status categories. Probiotic supplementation differentially influenced gut microbiota composition across study groups: *Butyricicoccus*, *Clostridium* sensu stricto 1, and *Coprococcus* were significantly enriched in the H group; the CCD group exhibited reduced abundances of *Haemophilus* and *Lachnospira*; *Anaerostipes* and *Fusicatenibacter* showed increases in the CMD group. Notably, we found that only long-term probiotic supplementation led to the enrichment of *Dorea*, *Eubacterium hallii* group, and *Blautia* in all three health status groups, while suppressing *Klebsiella* and *Bilophila* in the CMD and CCD groups. Enterotype analysis of both cohorts demonstrated significantly stronger microbiota modulation effects of probiotics in patients with CMD or CCD than in healthy individuals.

## Figures and Tables

**Figure 1 microorganisms-13-01507-f001:**
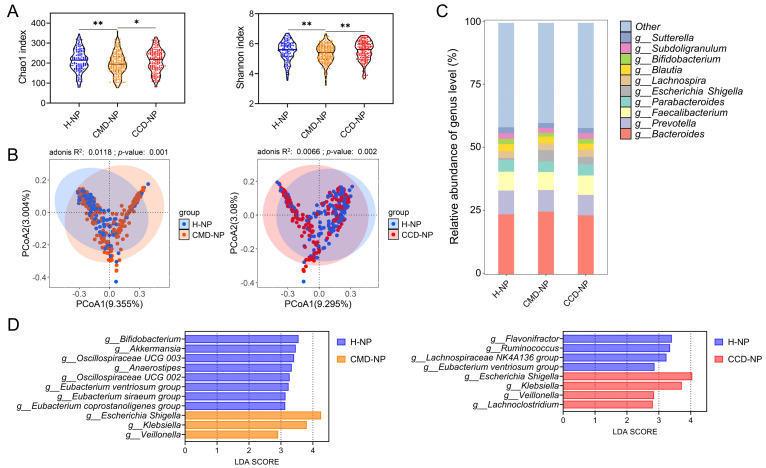
Differences in gut microbiota between the CMD and CCD groups and the healthy group in cohort 1, both without probiotic supplementation. (**A**) α-diversity assessed via Chao1 and Shannon indices at ASV level. (**B**) β-diversity evaluation of microbial communities at ASV level. Principal coordinate analysis (PCoA) visualization employing Bray–Curtis distance. Statistical evaluation of intergroup differences was implemented through adonis tests (999 iterations), with R^2^ values quantifying explained variance and *p*-values denoting significance displayed in panels. (**C**) Global composition of gut microbiota at the genus level within NP groups. Only the top 10 taxa are presented in the graph. (**D**) LEfSe comparisons between the CMD and CCD groups and the healthy group. Only taxonomic features with LDA scores exceeding 2.0 (log10 scale) were considered. This analytical approach was restricted to bacterial genera exhibiting relative abundances above 0.1%. Statistical significance was determined using Mann–Whitney U test, *, *p* < 0.05, **, *p* < 0.01.

**Figure 2 microorganisms-13-01507-f002:**
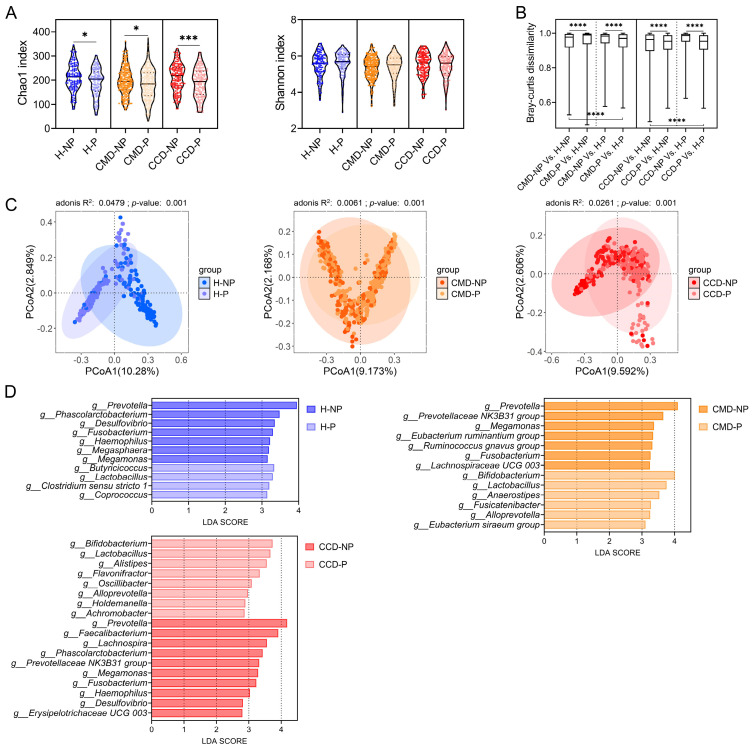
The effect of probiotic supplementation on the gut microbiota of three health status categories in cohort 1. (**A**) α-diversity assessed through Chao1 and Shannon indices at ASV level. (**B**) Comparative analysis of Bray–Curtis dissimilarity measures between the CMD and CCD groups and the healthy group, categorized according to probiotic supplementation status. (**C**) β-diversity evaluation of microbial communities at ASV level. PCoA visualization employing Bray–Curtis distance. Statistical evaluation of intergroup differences was implemented through adonis tests (999 iterations), with R^2^ values quantifying explained variance and *p*-values denoting significance displayed in panels. (**D**) LEfSe comparisons between P and NP groups within each health status category (CMD, CCD, and healthy groups). Only taxonomic features demonstrating LDA scores exceeding 2.0 (log10 scale) were considered. This analytical approach was restricted to bacterial genera exhibiting relative abundances above 0.1%. Statistical significance was determined using Mann–Whitney U test; *, *p* < 0.05; ***, *p* < 0.001; ****, *p* < 0.0001.

**Figure 3 microorganisms-13-01507-f003:**
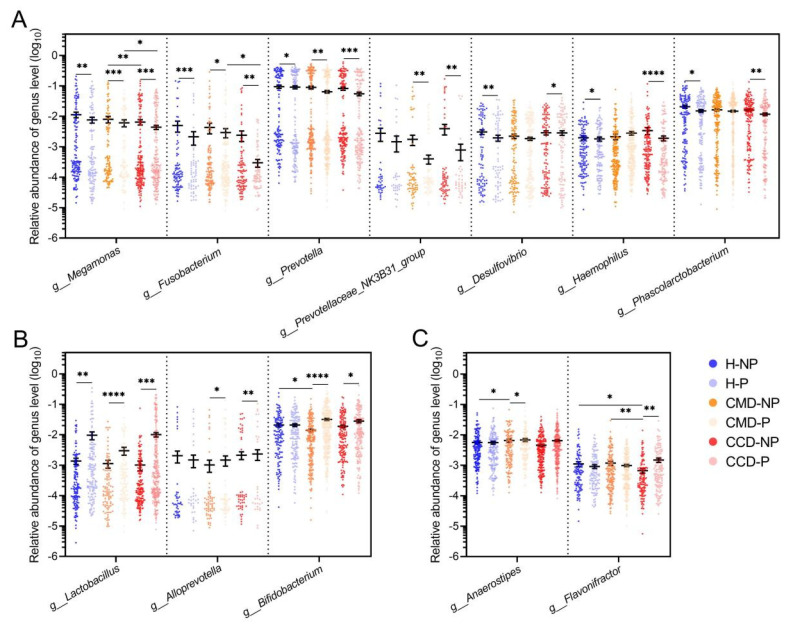
Changes in gut bacteria across groups in cohort 1 resulting from probiotic supplementation. (**A**) Gut bacteria inhibited by probiotic supplementation. Only bacteria with significantly decreased relative abundance in at least two P groups (including CCD-P, CMD-P, and H-P) compared to their corresponding NP groups (including CCD-NP, CMD-NP, and H-NP) are shown. (**B**) Gut bacteria promoted by probiotic supplementation. Only bacteria with significantly increased relative abundance in at least two P groups (including CCD-P, CMD-P, and H-P) compared to their corresponding NP groups (including CCD-NP, CMD-NP, and H-NP) are shown. (**C**) Gut microbes that exhibit the same trend of change in both the healthy group without probiotic supplementation (H-NP) and the CMD and CCD groups with probiotic supplementation (CMD-P and CCD-P), relative to the CMD and CCD groups without probiotic supplementation (CMD-NP and CCD-NP). Statistical significance was determined using Mann–Whitney U test; *, *p* < 0.05; **, *p* < 0.01; ***, *p* < 0.001; ****, *p* < 0.0001.

**Figure 4 microorganisms-13-01507-f004:**
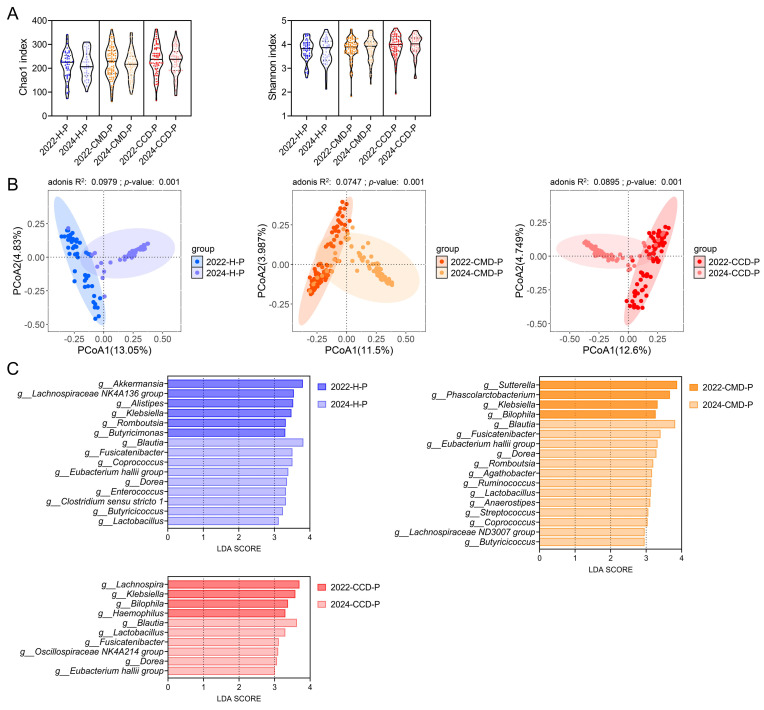
The effect of long-term probiotic supplementation on the gut microbiota of three health status categories in cohort 2. (**A**) α-diversity assessed through Chao1 and Shannon indices at ASV resolution. (**B**) β-diversity evaluation of microbial communities at ASV level. PCoA visualization employing Bray–Curtis distance. Statistical evaluation of intergroup differences was implemented through adonis tests (999 iterations), with R^2^ values quantifying explained variance and *p*-values denoting significance displayed in panels. (**C**) LEfSe comparisons between 2022 and 2024 batches within each health status category (CMD, CCD, and healthy groups). Taxonomic features demonstrating LDA scores exceeding 2.0 (log10 scale) were considered. This analytical approach was restricted to bacterial genera exhibiting relative abundances above 0.1%.

**Figure 5 microorganisms-13-01507-f005:**
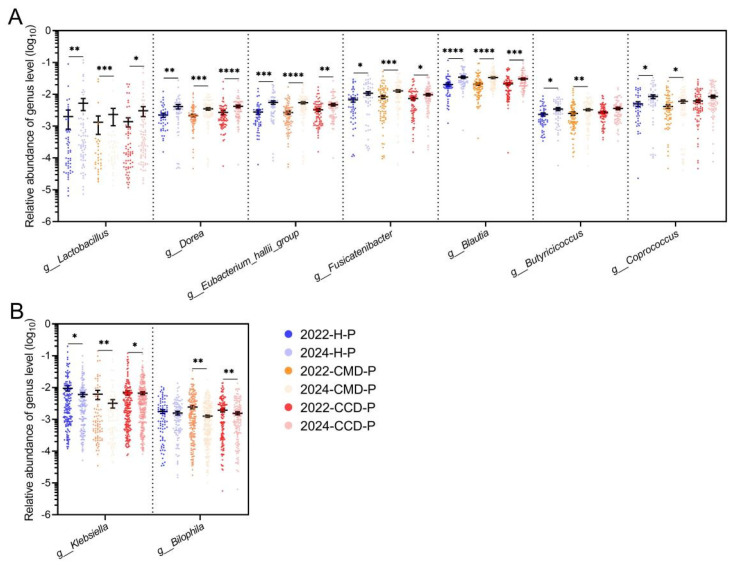
Changes in gut bacteria across groups in cohort 2 resulting from long-term probiotic supplementation. (**A**) Gut bacteria promoted by long-term probiotic supplementation. Only bacteria with significantly increased relative abundance in at least two of the health status groups within the 2024 batch (including 2024-CCD-P, 2024-CMD-P, and 2024-H-P) compared to their counterpart groups in the 2022 batch (including 2022-CCD-P, 2022-CMD-P, and 2022-H-P) are presented. (**B**) Gut bacteria inhibited by long-term probiotic supplementation. Only bacteria with significantly decreased relative abundance in at least two of the health status groups within the 2024 batch (including 2024-CCD-P, 2024-CMD-P, and 2024-H-P) compared to their counterpart groups in the 2022 batch (including 2022-CCD-P, 2022-CMD-P, and 2022-H-P) are presented. Mann–Whitney U test was used for statistical analysis; *, *p* < 0.05; **, *p* < 0.01; ***, *p* < 0.001; ****, *p* < 0.0001.

**Figure 6 microorganisms-13-01507-f006:**
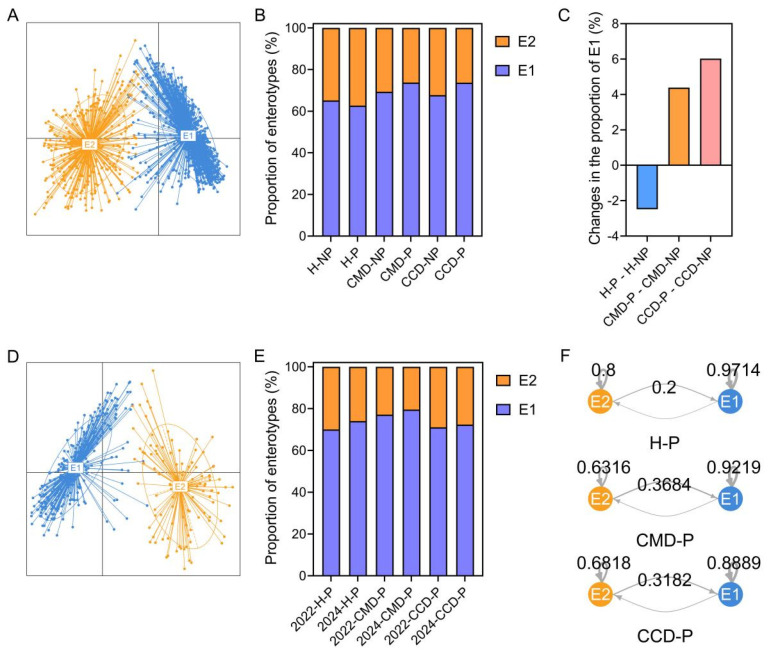
Effect of probiotic supplementation on enterotypes in individuals with different health status. (**A**) Enterotype analysis of cohort 1. (**B**) Distribution of enterotype proportions across groups in cohort 1. (**C**) Changes in E1 proportions between the P groups and their corresponding NP groups in cohort 1. (**D**) Enterotype analysis of cohort 2. (**E**) Distribution of enterotype proportions across groups in cohort 2. (**F**) Markov chain analysis of transition probabilities between the two enterotypes in cohort 2. Arrow weights represent the maximum likelihood estimates of the transition probabilities between different states. Only probabilities greater than 0.2 are displayed. E1, enterotype 1; E2, enterotype 2. The dominant contributing bacteria for enterotype 1 are *Bacteroides*, and those for enterotype 2 are *Prevotella*.

**Table 1 microorganisms-13-01507-t001:** Distribution of participants in cohort 1.

Characteristics	Total (%)	CMD (%)	CCD (%)	H (%)
	(*n* = 1168)	(*n* = 547)	(*n* = 339)	(*n* = 282)
Individuals with probiotic supplementation	660 (56.51)	335 (61.24)	175 (51.62)	150 (53.19)
Individuals without probiotic supplementation	508 (43.49)	212 (38.76)	164 (48.38)	132 (46.81)

**Table 2 microorganisms-13-01507-t002:** Sample characteristics in cohort 2.

Characteristics	Total	CMD-P	CCD-P	H-P
	(*n* = 418)	(*n* = 166)	(*n* = 152)	(*n* = 100)
Sample collection				
2022 (October–December)	209	83	76	50
2024 (January–March)	209	83	76	50

## Data Availability

Raw Sequencing data have been deposited to the NCBI SRA project under the NCBI BioProject ID PRJNA1261611, PRJNA1261608, and PRJNA1261599.

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
