# Peer review of "Probiotic Supplementation Improves Gut Microbiota in Chronic Metabolic and Cardio-Cerebrovascular Diseases Among Chinese Adults over 60: Study Using Cross-Sectional and Longitudinal Cohorts"

_microorganisms, 2025, doi:10.3390/microorganisms13071507_

Round 1
Reviewer 1 Report
Comments and Suggestions for Authors
microorganisms-3695450-peer-review-v1
Paper is interesting and provide some information regarding role of the use of long term applications of probiotics with focus of Asian ethnic background. In my opinion paper deserve attention form the editor, however, some minor adjustments and corrections from the authors will need to be taken into account.
The title will need to be corrected. Since 2020 the taxonomy of lactobacilli has changed. In the paper authors will need to adjust taxonomic names and apply recommended abbreviations (doi: 10.1163/18762891-20230114)later into the text. According to paper introduced the new taxonomy in 2020 (Zheng et al., 2020 doi: 10.1099/ijsem.0.004107) when referring to genus Lactobacillus in sense of term before 2020, English word "lactobacilli" needs to be used. In this content, in the title it will be more appropriate to say "... lactobacilli and bifidobacteria....". Please, similar adjustments needs to be considered in the entire manuscript.
Ln25: please, see my previous comment
Ln41: Please pay attention to the format of the manuscript. Add intervale before references. Please, check the entire manuscript for similar adjustments.
Ln48: It is really important that this specific strain (Lachnoclostridium saccharolyticu WM1) is proliferating? Maybe use of strain identification (WM1) is not really needed in this context.
Ln72: Please, strain identification (BBG-001) do not need to be in italics. Please, check the entire manuscript for similar adjustments.
Introduction is well structured and presents principal points regarding beneficial properties of probiotics and other alternatives, some questionable sides and challenges faced by the scientific community.
Ln88: Please, adjust names of the applied probiotics according to new taxonomy; subsp. do not need to be in italics.
Please for all suppliers of the company with equipment for the name of the company link to appropriate addresses, including city, state (in case of federal country) in abbreviated way, and name of the country. Please, on the following occasions, only the name of the company is sufficient. When providing this information, try to refer to the headquarters of the company, and not to local distributors.
On Ln113 was mentioned that Ethical approval was from Chines Academy of Science, however, on Ln619 this was Peking University. With the aim of transparency, authors will need to provide a copy of the certificate.
Ln135: Working with DNA can maybe give some results that may have double interpretation. DNA can be obtained from dead cells, and the interpretation of the results can be questionable. Are authors performed any additional steps/analyses with aim to do interpretation of the results and analyze only DNA from life cells? Some pre-treatments of the samples can be option; or work with RNA. Please, can you comment on this specific point in the material and methods and later in results and discussion.
Ln340: B from Butyricimonas, needs to be in italics.
In the discission section it will be appropriate to have some additional references. There are several very long sections of texts without references. Maybe this point can be taken into account by authors.
Author Response
Response to Reviewer 1 Comments:
Comments 1. The title will need to be corrected. Since 2020 the taxonomy of lactobacilli has changed. In the paper authors will need to adjust taxonomic names and apply recommended abbreviations (doi: 10.1163/18762891-20230114) later into the text. According to paper introduced the new taxonomy in 2020 (Zheng et al., 2020 doi: 10.1099/ijsem.0.004107) when referring to genus Lactobacillus in sense of term before 2020, English word "lactobacilli" needs to be used. In this content, in the title it will be more appropriate to say "... lactobacilli and bifidobacteria....". Please, similar adjustments needs to be considered in the entire manuscript.
Response: Thank you for your comments. Following the updated taxonomy since 2020, all mentions of “Lactobacillus” and “Bifidobacterium” have been revised to “lactobacilli” and “bifidobacteria” throughout the manuscript (with the exception of specific species names), with detailed explanations added in the “Materials and Methods” section and supporting references included (Reference: Jinshui, Z. et al. A taxonomic note on the genus Lactobacillus: Description of 23 novel genera, emended description of the genus Lactobacillus Beijerinck 1901, and union of Lactobacillaceae and Leuconostocaceae. Int J Syst Evol Microbiol (2020); Todorov, S D. et al. Recommendations for the use of standardised abbreviations for the former Lactobacillus genera, reclassified in the year 2020. Benef Microbes (2023)). Please see lines 172 to 174.
Additionally, all bacterial strains cited in this study have been verified and updated to their current scientific names using the LPSN and NCBI Taxonomy databases, as noted in the manuscript. Please see lines 95 to 98.
Comments 2. Ln25: please, see my previous comment.
Response: Thank you for your suggestions. The changes have been made according to the recommendations in “Comments 1”. Please see line 26.
Comments 3. Ln41: Please pay attention to the format of the manuscript. Add intervale before references. Please, check the entire manuscript for similar adjustments.
Response: Thank you for your suggestions. Appropriate line spacing has been inserted preceding the reference section, and a comprehensive review of manuscript formatting has been conducted to ensure uniformity.
Comments 4. Ln48: It is really important that this specific strain (Lachnoclostridium saccharolyticum WM1) is proliferating? Maybe use of strain identification (WM1) is not really needed in this context.
Response: Thank you for your suggestions. We sincerely appreciate this valuable input. The strain identifier (WM1) has been removed from "Lachnoclostridium saccharolyticum" in the revised text, aligning with your recommendation for contextual relevance. Please see line 48.
Comments 5. Ln72: Please, strain identification (BBG-001) do not need to be in italics. Please, check the entire manuscript for similar adjustments.
Response: Thank you for your comments. All concerns regarding microbial nomenclature formatting conventions (including strain designations) have been comprehensively addressed in accordance with standardized taxonomic guidelines, with particular attention to italicization protocols. Please see line 75.
Comments 6. Introduction is well structured and presents principal points regarding beneficial properties of probiotics and other alternatives, some questionable sides and challenges faced by the scientific community.
Response: Thank you for your positive feedback. We have further strengthened the introduction's logical rigor by adding a description of the pathological similarities between chronic metabolic disease and cardio-cerebrovascular disease, as well as gut microbiota characteristics in older adults, to highlight the rationale for sample selection. Please see lines 50 to 53 and lines 79 to 86.
Comments 7. Ln88: Please, adjust names of the applied probiotics according to new taxonomy; subsp. do not need to be in italics.
Response: Thank you for your comments. All concerns regarding microbial nomenclature formatting conventions (including strain designations) have been comprehensively addressed in accordance with standardized taxonomic guidelines, with particular attention to italicization protocols. Please see line 96.
Comments 8. Please for all suppliers of the company with equipment for the name of the company link to appropriate addresses, including city, state (in case of federal country) in abbreviated way, and name of the country. Please, on the following occasions, only the name of the company is sufficient. When providing this information, try to refer to the headquarters of the company, and not to local distributors.
Response: Thank you for your comments. Yingdong Intelligent Technology (Shandong) Co., Ltd. is the official corporate headquarters. The company address has been supplemented in the “Materials and Methods” section per your recommendation. Please see line 102.
Comments 9. On Ln113 was mentioned that Ethical approval was from Chines Academy of Science, however, on Ln619 this was Peking University. With the aim of transparency, authors will need to provide a copy of the certificate.
Response: We really appreciate your in-depth comments. We sincerely apologize for the editorial error in the ethical approval statement. As correctly noted by the reviewer, the ethical approval for this study was indeed granted by the Ethical Review Committee of Peking University (Approval No. IRB00001052-18039, dated 18 June 2018), which has been properly documented in the Institutional Review Board Statement section. Please see lines 655 to 657. The mention of 'Chinese Academy of Science' in the “Materials and Methods” section resulted from an inadvertent copy-paste remnant carried over from a previous document template. Regarding the request for a copy of the ethics certificate, we have provided photographs of the ethics approval documents.
It is important to specifically note that Jinan Fuguo Tianrui Technology Co., Ltd., due to equity restructuring, completed its industrial and commercial deregistration on November 18, 2019, and was renamed Yingdong Intelligent Technology (Shandong) Co., Ltd. The health management business of the former company has been fully inherited by the latter. Relevant information is publicly available for inquiry.

Comments 10. Ln135: Working with DNA can maybe give some results that may have double interpretation. DNA can be obtained from dead cells, and the interpretation of the results can be questionable. Are authors performed any additional steps/analyses with aim to do interpretation of the results and analyze only DNA from life cells? Some pre-treatments of the samples can be option; or work with RNA. Please, can you comment on this specific point in the material and methods and later in results and discussion.
Response: We sincerely appreciate the reviewer's valuable feedback on addressing the potential ambiguity of DNA analysis from dead bacteria. Below, we provide a point-by-point response:
DNA Extraction Methodology: Our study utilized the QIAamp PowerFecal DNA Kit (Qiagen), which employs a combination of mechanical (0.1 mm grinding beads) and chemical lysis (Buffer GS) to isolate DNA. The silica membrane adsorption step specifically targets intact DNA fragments (200 bp–50 kb), which partially filters highly fragmented DNA from old dead bacteria (>72 hours post-mortem). However, DNA from recently deceased bacteria (<24 hours), which retain longer fragments due to partial cellular integrity, may still be captured. This limitation is inherent to standard DNA extraction protocols and aligns with methodologies widely adopted in high-impact studies (Reference: Aikun, F. et al. Tumor-resident intracellular microbiota promotes metastatic colonization in breast cancer. Cell (2022)).
Mitigating False Positives in 16S rRNA Sequencing: While dead bacteria may contribute DNA, sequence misclassification (the primary cause of false positives) is minimized by the DADA2 denoising algorithm (Reference: Benjamin, J. C. et al. DADA2: High-resolution sample inference from Illumina amplicon data. Nature Methods (2016)). DADA2 reduces spurious reads by clustering highly similar sequences, thereby improving taxonomic resolution. Although it cannot distinguish the physiological state of DNA, this step inherently limits confounding effects from fragmented or low-quality DNA.
Study Scope and Implications: Our primary goal was to assess overall shifts in gut microbiota composition following probiotic intervention, rather than tracking viable/culturable bacteria. We acknowledge that extracellular DNA from dead bacteria inherently remains part of the gut microbiome’s ecological "footprint." Including such DNA in diversity analysis does not substantially alter our core findings, as structural changes in microbial communities (viable or not) remain mechanistically relevant to host health outcomes.
Future Directions: We fully agree that distinguishing live/dead bacterial DNA poses a critical challenge. Future studies integrating RNA analysis, viability-selective culturing, or advanced techniques like propidium monoazide (PMA) pretreatment may further refine these assessments.
Comments 11. Ln340: B from Butyricimonas, needs to be in italics.
Response: Thank you for your meticulous review. The genus name "Butyricimonas" have been italicized in accordance with taxonomic formatting standards. Please see line 360.
Comments 12. In the discission section it will be appropriate to have some additional references. There are several very long sections of texts without references. Maybe this point can be taken into account by authors.
Response: Thank you for this thoughtful suggestion. We have carefully revised the Discussion section to incorporate additional references in areas where supporting literature was previously sparse. Citations have now been added to lines 469, 472, 473, 483, 491, 502, 508, 512, 564 and 566 to better contextualize our findings.

Reviewer 2 Report
Comments and Suggestions for Authors
This study presents a robust analysis, supported by a large and well-characterized cohort, of the effect of probiotic supplementation in modulating the gut microbiota in elderly individuals (>60 years) with chronic metabolic disease (CMD) or cardio-cerebrovascular disease (CCD), compared with healthy subjects. Both cross-sectional and longitudinal approaches were employed, integrating analyses of diversity, taxonomic abundance, and enterotipic transition. Results suggest that probiotic supplementation induces significant changes in microbial composition, promotes potentially beneficial genera, and reduces pathobionts in disease groups. Furthermore, the effect is greater in subjects with pre-existing dysbiosis. The study is well written, methodologically sound and clearly structured. However, there are several critical aspects that need attention or clarification.
- The title is overly general and descriptive. Suggestion: make it more specific to indicate the target cohort (age >60, Chinese population) and design (cross-sectional and longitudinal cohorts). Also, I would tend not to use abbreviation in the title.
- In general, in the text always lacks space between the last letter of a word and the square bracket relating to the subsequent reference. Please correct.
- Direct references to the prevalence or specificity of the microbiota in Chinese elderly subjects are lacking. It is important to better justify the choice of this population.
- The distinction between CMD and CCD is treated as clear cut, but many patients have comorbidities. This aspect should be mentioned.
- Strains are listed, but concentration (CFU per dose) and daily dosing are missing.
- The definition of CMD and CCD is self-reported (“self-reported”) but it is unclear whether there was clinical verification (e.g., medical records).
- Mean time (13.77 months) is reported, but variability is absent. What is the minimum and maximum range?
- The number of samples compared in each group varies significantly: was sample balancing considered?
- It is unclear what group H individuals comprise.
- In FIG.1 A the statistical differences are not very clear as to which groups they are related to, also because in the text it is reported that the H group has a difference only with the CCD group, so why are there 2 asterisks?
- Fig 1 D, it is not clear why in the two images the bacterial families relating to group H are different from each other.
- Line 246/253 - only one p-value is reported, even though 2 are shown in the image, please correct.
- It is unclear why the microbiota of H-group individuals varies so greatly after probiotic use despite precisely being healthy individuals.
- Prevotella reduction is presented as beneficial. However, Prevotella may be neutral or beneficial depending on the dietary context. More caution would be needed.
- Line 490 has the abbreviation CVD for cardiovascular disease notwithstanding that this phrase has been used before (Line 437)
- The enterotype analysis is interesting, but the interpretation of E2 to E1 transitions requires more caution, considering that the literature is not unambiguous about the “superiority” of enterotype 1.
- References to possible side effects or tolerability of probiotics are lacking.
- Some obvious limitations are not adequately discussed: e.g., nonrandomized design, risk of residual confounding (diet, medications, intercurrent diseases).
- Any mention of participants' dietary composition, which can heavily influence the microbiota, is missing.
- What was the level of adherence to probiotic supplementation in longitudinal subjects?
- Were data collected on diet, concomitant medications (e.g., statins, antidiabetics), or other confounding factors?
Author Response
Response to Reviewer 2 Comments:
Comments 1. The title is overly general and descriptive. Suggestion: make it more specific to indicate the target cohort (age >60, Chinese population) and design (cross-sectional and longitudinal cohorts). Also, I would tend not to use abbreviation in the title.
Response: We appreciate the reviewer’ s constructive feedback. The title has been revised to "Probiotic Supplementation Improves Anaerostipes and Fusicatenibacter in Chronic Metabolic Diseases Among Chinese Adults Over 60 Through Cross-Sectional and Longitudinal Cohorts”.
Comments 2. In general, in the text always lacks space between the last letter of a word and the square bracket relating to the subsequent reference. Please correct.
Response: Thank you for your suggestions. Appropriate line spacing has been inserted preceding the reference section, and a comprehensive review of manuscript formatting has been conducted to ensure uniformity.
Comments 3. Direct references to the prevalence or specificity of the microbiota in Chinese elderly subjects are lacking. It is important to better justify the choice of this population.
Response: Thank you for your comments.We revised the “Introduction” section to further elaborate on the characteristics of gut microbiota in elderly individuals, clarifying our rationale for selecting this population. Please see lines 79 to 86.
Comments 4. The distinction between CMD and CCD is treated as clear cut, but many patients have comorbidities. This aspect should be mentioned.
Response: Thank you for your comments. In response to your suggestions, we have further described the similar characteristics between CMD and CCD, as well as the onset of comorbidities. Please see lines 50 to 53.
Comments 5. Strains are listed, but concentration (CFU per dose) and daily dosing are missing.
Response: We appreciate this important clarification request. As detailed in Section 2.1 ("Probiotic Strains"), participants consumed Basegar® probiotic containing Lacticaseibacillus paracasei Lpc-37, Lacticaseibacillus rhamnosus HN001, Bifidobacterium ani-malis subsp. lactis HN019, Lactobacillus acidophilus NCFM, and Lacticaseibacillus rhamnosus Lr-32. Regrettably, the CFU concentrations remain proprietary information not publicly disclosed by the manufacturer. Regarding daily dosing, participants voluntarily administered the probiotic without clinical supervision. We acknowledge this limitation in “Materials and Methods” section. Please see lines 97 to 98 and lines 124 to 126.
Comments 6. The definition of CMD and CCD is self-reported (“self-reported”) but it is unclear whether there was clinical verification (e.g., medical records).
Response: We appreciate your valid methodological concern. Regarding clinical verification, we acknowledge that access to personal medical records was not provided by participants due to privacy protections. CMD and CCD definitions were established through self-reported conditions, supplemented by medication types (excluding dosage and duration) for validation. This content is further elaborated in the “Materials and Methods” section. Please see lines 110 to 113 and lines 116 to 118.
Comments 7. Mean time (13.77 months) is reported, but variability is absent. What is the minimum and maximum range?
Response: We thank you for flagging this statistical clarification. The long-term probiotic usage group showed intervals of 12 months (n=12), 13 months (n=28), 14 months (n=167), 15 months (n=1), and 16 months (n=1), yielding the 13.77-month mean, with 95% of participants clustered within the 12–14-month range and a median of 14 months. This content is further elaborated in the “Materials and Methods” section. Please see lines 148 to 150.
Comments 8. The number of samples compared in each group varies significantly: was sample balancing considered?
Response: We sincerely appreciate the reviewer’s insightful comment regarding the sample size imbalance across groups. The sample size disparity (particularly the larger cohort in the CMD group compared to the CCD and H groups) stems from the scarcity of available samples. To maximize statistical power while preserving biological relevance, we prioritized retaining all qualified samples rather than artificially balancing group sizes.
We acknowledge that unequal sample sizes may introduce potential bias and have addressed this concern through the following methodological precautions: Non-parametric tests (Mann-Whitney U test) were systematically employed for α-diversity (Figs. 1A, 2A), genus-level abundance comparisons (Figs. 3, 5), and LEfSe-based (Kruskal-Wallis test) inter-group differential bacterial analysis (Figs. 2D, 4C), as these methods are less sensitive to unequal sample sizes compared to parametric tests. The β-diversity assessment employed a weighted PERMANOVA (Adonis) analysis with 999 permutations (Figure 2B) to mitigate the impact of sample balance issues.
We recognize that even with these precautions, sample imbalance could theoretically influence microbial diversity metrics. This limitation has been explicitly acknowledged in the revised Discussion section (lines 617–618). Future multi-center studies with balanced prospective sampling will further validate these findings.
Comments 9. It is unclear what group H individuals comprise.
Response: We appreciate the opportunity to clarify this point. The H group (healthy controls) comprised individuals who self-reported no major chronic conditions, with adjustments made for recent medication usage as detailed in the “Materials and Methods” section. Please see lines 118 to 122.
Comments 10. In FIG.1 A the statistical differences are not very clear as to which groups they are related to, also because in the text it is reported that the H group has a difference only with the CCD group, so why are there 2 asterisks?
Response: We appreciate your inquiry. The two asterisks in Figure 1A correspond to separate comparisons: one reflects the alpha-diversity difference between the H and CCD groups (as noted), while the other represents the comparative analysis between the CMD and CCD groups. As mentioned in lines 226-230 of the Results section, the alpha-diversity of the CCD-NP and H-NP groups was significantly higher than that of the CMD-NP group. Additionally, lines 364-366 describe that no significant alpha-diversity differences were observed between the CCD-NP and H-NP groups.
Comments 11. Fig 1 D, it is not clear why in the two images the bacterial families relating to group H are different from each other.
Response: Thank you for your thoughtful observation. Regarding the differing bacterial families associated with the H group observed in the gut microbiota differential analysis of Figure 1D, we humbly speculate that this discrepancy might be attributed to the CCD group encompassing a wide variety of disease types but having a relatively limited sample size. We believe that expanding the sample size might improve these results. The CMD and CCD groups are both collections formed through combinations of multiple diseases, with the CMD group including 4 diseases (CMD-NP, 212 individuals) and the CCD group containing 11 diseases (CCD-NP, 164 individuals). Through LEfSe analysis conducted between each disease group and the healthy group, we found that only the Eubacterium ventriosum group was shared among the bacteria associated with the H group. However, in contrast, the bacteria associated with both disease groups were highly similar. Although sample size limitations prevented us from identifying more precise bacterial biomarkers, the current findings sufficiently demonstrate significant differences in gut microbiota between both disease groups and the healthy control group, while also providing a foundation for subsequent characterization of how probiotics influence gut microbiota in populations with differing health status.
Comments 12. Line 246/253 - only one p-value is reported, even though 2 are shown in the image, please correct.
Response: Thank you for your suggestion. The p-values have been supplemented in the Results section. Please see line 275.
Comments 13. It is unclear why the microbiota of H-group individuals varies so greatly after probiotic use despite precisely being healthy individuals.
Response: We appreciate this thoughtful observation. Although healthy individuals typically exhibit relatively stable microbiota, there is evidence suggesting that even in non-diseased populations, responses to probiotics may vary significantly, potentially influenced by unmeasured lifestyle factors (Jan, M. et al. Lifestyle interventions to delay senescence. Biomed J (2023)) or other individual differences (Javad, S. R. et al. Probiotics: Versatile Bioactive Components in Promoting Human Health. Medicina (2020)). On another note, the unique gut microbiota structure in elderly populations may more readily manifest probiotic effects. Analysis combining two cohorts revealed that probiotic supplementation exerted greater influence on gut microbiota in both CMD and CCD disease groups compared to healthy populations. Future studies utilizing larger cohorts and multi-omics analyses will help elucidate variations in probiotic effects on healthy populations.
Comments 14. Prevotella reduction is presented as beneficial. However, Prevotella may be neutral or beneficial depending on the dietary context. More caution would be needed.
Response: We agree that Prevotella’s ecological role is highly context-dependent (Abdelsalam, N. A. et al. The curious case of Prevotella copri. Gut Microbes (2023)), particularly in relation to dietary patterns such as high-fiber diets where it may act as a commensal or even beneficial taxon (Chang, H. W. et al. Prevotella copri and microbiota members mediate the beneficial effects of a therapeutic food for malnutrition. Nat Microbiol (2024)). In our study, the reduction of Prevotella was interpreted as potentially beneficial. However, this study did not conduct bacterial-related functional validation and was unable to directly assess the role of Prevotella. We adjusted our discussion to avoid overgeneralization. Please see lines 564-568.
Comments 15. Line 490 has the abbreviation CVD for cardiovascular disease notwithstanding that this phrase has been used before (Line 437)
Response: Thank you for your meticulous correction regarding abbreviation standards. We have removed the “CVD” abbreviation in line 515 (as this term is not central to the study) and consistently used the full term “cardiovascular disease” throughout the text to avoid redundancy. All related expressions in the manuscript have been revised accordingly.
Comments 16. The enterotype analysis is interesting, but the interpretation of E2 to E1 transitions requires more caution, considering that the literature is not unambiguous about the “superiority” of enterotype 1.
Response: Thank you for highlighting the need for caution in interpreting enterotype transitions. We have revised the speculation in the Discussion section regarding the potential health benefits of the transition from E2 to E1. Please see lines 585-588.
Comments 17. References to possible side effects or tolerability of probiotics are lacking.
Response: Thank you for your suggestion. This paper mentions the existing side effects of probiotics in the introduction (please see lines 76-78) and further supplements the description of potential side effects associated with probiotic supplementation in the discussion section (please see lines 467-473).
Comments 18. Some obvious limitations are not adequately discussed: e.g., nonrandomized design, risk of residual confounding (diet, medications, intercurrent diseases).
Response: Thank you for your feedback. As an observational study, this research indeed carries possible residual confounding factors. We have supplemented the Discussion section (please see lines 612-617) with an explanation of the limitations inherent to the non-randomized design and proposed that future studies reduce confounding by utilizing more detailed dietary surveys and medication registries.
Comments 19. Any mention of participants' dietary composition, which can heavily influence the microbiota, is missing.
Response: We fully recognize the importance of dietary composition. Due to data collection limitations, the study failed to include detailed dietary components and quantitative information, which has been explicitly stated in the limitations section of the discussion (please see lines 616-617). Future study designs will prioritize the integration of standardized dietary records to enable a more comprehensive assessment of the impact of dietary composition on gut microbiota.
Comments 20. What was the level of adherence to probiotic supplementation in longitudinal subjects?
Response: Thank you for the insightful question regarding adherence monitoring. Since the screening of longitudinal study participants was based on long-term probiotic purchases, we acknowledge inherent limitations: actual intake adherence (daily dosage) cannot be directly verified beyond probiotic purchase intervals. We have supplemented the limitations section in the Discussion (please see lines 612-616). Subsequent studies need to establish more comprehensive monitoring models.
Comments 21. Were data collected on diet, concomitant medications (e.g., statins, antidiabetics), or other confounding factors?
Response: Thank you for raising the question regarding confounding factors. Regarding confounding factor data, we only recorded major drug categories and basic dietary patterns (e.g., vegetarian/omnivorous), but did not systematically collect dosage and temporal information. This was emphasized in the Discussion section (please see lines 612-614) as a limitation affecting the interpretation of results.

Round 2
Reviewer 2 Report
Comments and Suggestions for Authors
The authors clearly and thoroughly answered all questions.